# Efficient RO-PUF for Generation of Identifiers and Keys in Resource-Constrained Embedded Systems

**Macarena C. Martínez-Rodríguez** [ID], **Luis F. Rojas-Muñoz** [ID], **Eros Camacho-Ruiz** [ID], **Santiago Sánchez-Solano \*** [ID] **and Piedad Brox** [ID]

Instituto de Microelectrónica de Sevilla, IMSE-CNM, CSIC/Universidad de Sevilla, 41092 Sevilla, Spain
\* Correspondence: santiago@imse-cnm.csic.es; Tel.: +34-954466666

**Abstract:** The generation of unique identifiers extracted from the physical characteristics of the underlying hardware ensures the protection of electronic devices against counterfeiting and provides security to the data they store and process. This work describes the design of an efficient Physical Unclonable Function (PUF) based on the differences in the frequency of Ring Oscillators (ROs) with identical layout due to variations in the technological processes involved in the manufacture of the integrated circuit. The logic resources available in the Xilinx Series-7 programmable devices are exploited in the design to make it more compact and achieve an optimal bit-per-area rate. On the other hand, the design parameters can also be adjusted to provide a high bit-per-time rate for a particular target device. The PUF has been encapsulated as a configurable Intellectual Property (IP) module, providing it with an AXI4-Lite interface to ease its incorporation into embedded systems in combination with soft- or hard-core implementations of general-purpose processors. The capability of the proposed RO-PUF to generate implementation-dependent identifiers has been extensively tested, using a series of metrics to evaluate its reliability and robustness for different configuration options. Finally, in order to demonstrate its utility to improve system security, the identifiers provided by RO-PUFs implemented on different devices have been used in a Helper Data Algorithm (HDA) to obfuscate and retrieve a secret key.

**Keywords:** hardware security; physical unclonable functions; device authentication; key generation; reconfigurable devices; embedded systems

## 1. Introduction

The combination of encryption, authentication, and data verification provides robust and reliable mechanisms necessary to guarantee the security of the information captured, processed, and transmitted by devices today connected to the Internet to support a multitude of services related to leisure, health, business, or industry [1,2]. To be truly effective, security protocols for the authentication and integrity of critical data need to be grounded in hardware and not reliant on pure software-based solutions. The grounding of security in silicon manufacturing processes provides a hardened layer of protection that increases confidence in electronic systems [3,4].

Physically Unclonable Functions (PUFs) have emerged as a potential solution to build trusted anchors that provide secure hardware solutions for consumer and industrial IoT devices [5]. Based on their properties, PUFs can be used to generate unique identifiers that facilitate device authentication to prevent spoofing and counterfeiting. They also introduce an extra hardware-based layer for building lightweight encryption schemes, as they can be used to obfuscate the secret keys used by ciphers, ensuring the confidentiality of data exchanged by the electronic device in which the PUF is embedded or attached. In addition, PUFs can provide seeds to be used in the creation of public and private key pairs for public-key cryptography, increasing system security by avoiding the need to share secret keys.

A PUF maps an input challenges sequence to an output response in a unique (it cannot be replicated, that is, it is unclonable), reliable (it can be reproducible over time), and unpredictable (it cannot be anticipated) way. The initial idea for this type of one-way function was introduced by Pappu et al. in [6]. The operating principle of silicon PUFs is based on the variations that arise during the manufacturing process of an integrated circuit. Roughly, research on silicon PUFs has focused on three categories: (i) memory-based PUFs (SRAM [7], DRAM [8,9]) that use unpredictable start-up values of memory cells; (ii) delay-based PUFs that use the relative time-delay differences between two theoretically identical circuits (Ring oscillators [10–19], Arbiter [20], Butterfly [21]); and (iii) analog PUFs that exploit measurements of variables in mixed-signal and analog integrated circuits (for instance, current mirrors [22]).

The use of Field-Programmable Gate Arrays (FPGAs) and programmable Systems on Chip (SoCs) to implement embedded systems for specific applications has experienced a significant boom in recent years. The possibility of incorporating general-purpose processors such as soft-cores in the former, or of using the powerful processing systems available in the latter, makes these reconfigurable devices very advantageous for providing solutions with reduced size, energy consumption, and cost, especially suitable for IoT applications. Security requirements for these implementations are identical to those for realizations employing Application-Specific Integrated Circuits (ASICs), so the proposed solutions are largely independent of the implementation technique.

PUF implementations on FPGAs have been mainly focused on RO-PUFs, since SRAM-based PUFs are not feasible, because the on-chip memories of programmable devices are usually initialized to a fixed value after start-up, and arbiter PUFs impose severe restrictions in the layout in order to obtain symmetric delay paths, which is difficult to achieve on programmable devices. RO-PUFs are based on closed delay chains (delay loops) whose oscillation frequencies are compared to obtain the PUF output. In theory, the oscillation frequencies of two ideally identical inverter chains should be the same, but this is not the case due to variability in the manufacturing process of the CMOS ICs that affects each device differently.

This paper describes the design of an RO-PUF to improve the security of embedded systems implemented on programmable devices. The combination of different design strategies proposed in the literature, as well as previous results obtained by the authors, gives rise to an efficient implementation in terms of resource consumption and speed of operation on Xilinx Zynq-7000 SoC devices. The design has been conceived as a configurable IP module, which includes mechanisms for the generation of the challenges sequence and the selection of the output bits, and it provides a standard connection interface to facilitate its incorporation as a basic block for the identification and generation of cryptographic keys in embedded systems. The paper also presents the different metrics used to verify the functionality of the PUF and perform its characterization under different operating conditions. The main contributions of the work are:

- Take advantage of the internal structure of programmable devices with the aim of obtaining a compact implementation of an RO-PUF. This strategy is carried out by using absolute and relative location directives in the VHDL descriptions used as input to the Vivado design tools for synthesis and implementation on Series-7 Xilinx devices.
- Include a mechanism for challenges generation that allows two different comparison strategies to be performed simultaneously. This feature doubles the number of bits generated per each comparison, maximizing the ratio between the length of the PUF response and the amount of resources required to obtain it.
- Implement the RO-PUF as a configurable IP module that can be connected to soft- and hard-core processors using standard interconnection buses. The size and placement of the RO bank in the programmable logic, as well as other design parameters that affect the performance of the PUF, can be selected by the designer through a graphical user interface supported by Vivado's IP Integrator tool.

- Provide a set of drivers that facilitates the use of the RO-PUF in a high-level programming language, such as C, and allows the development of software applications to calculate different metrics in order to evaluate its performance.

The paper is structured as follows: Section 2 provides a general review of the different RO-PUF designs available in the literature and combines the ideas coming from these sources with those obtained from previous works by the authors to define the specifications of a new RO-PUF whose architecture, building blocks, and use as an IP module are detailed in Section 3. The results obtained from the test battery used to verify the functionality and evaluate the performance of the PUF with different configuration options and operating conditions are collected and discussed in Section 4. Section 5 illustrates the use of the proposed RO-PUF as a basic element of an HDA to obfuscate and retrieve secret keys. Finally, the main conclusions obtained in this work are summarized in Section 6.

## 2. Ring Oscillator PUFs

A Ring Oscillator PUF (or RO-PUF) is a particular type of delay-based PUF whose operation is founded on the difference of frequencies in closed chains (rings) with an odd number of inverters. In practice, the ring usually consists of an even number of inverters and a NAND gate that receives an enable signal to open or close the feedback loop. When the loop is closed, each inverter generates an oscillating signal at its output, the frequency of which depends on the delays accumulated in the different stages and connection paths in the ring. Thus, two ROs with the same number of stages and identical layout should provide the same oscillation frequency. However, the frequencies are not equal because of the variability caused by the manufacturing processes of CMOS-integrated circuits, making each RO have a unique characteristic frequency.

In order to provide the appropriate number of bits to generate an identifier or obfuscate a cryptographic key, an RO-PUF must include a sufficient number of pairs of ROs. The RO-PUF proposed in [10], shown in Figure 1a, uses a bank of $N$ ROs that can be selected by pairs using two multiplexers. The concatenation of the signals that select the pairs of ROs to be compared, *challenge*1 and *challenge*2, allows establishing the sequence of challenges in this type of PUF. Each time a pair of ROs is selected, the frequencies of the two ROs are compared by connecting their outputs to two counters that will increase at the frequency determined by each RO. After a certain comparison time fixed externally, the counter values are compared to obtain a single bit response (so-called herein as "sign bit") depending on the counter that reaches a larger value. Due to each pair of ROs only generating one single-bit response, and only $N/2$ pairs of ROs being selected to provide non-correlated output, the generation of bitstreams with a large number of bits requires implementations with a large number of ROs.

This handicap is partially alleviated in the RO-PUF proposed in [17,18], which allows more than one bit to be added to the PUF response for each comparison. Unlike the conventional proposal in which the counting interval is fixed by an external clock, in this case, the decision is taken when the counter of the faster RO overflows. Then, the counter associated to the slower RO is stopped, and the response is taken from the output bits of this counter. For the selection of the response bits, the authors analyzed the entropy and the average probability and stability per bit, selecting those that provide the highest entropy and average stability while keeping the average probability around 50%. Each RO is used only once to generate the PUF response, splitting the $N$ ROs into two banks of $N/2$ ROs, where the *challenge* signal selects one RO per bank, as illustrated in Figure 1b.

To further increase the PUF response length for a given RO bank size, it is necessary to maximize the number of comparisons without compromising the PUF output bit correlation [13]. Area efficiency can also be improved resorting to the use of dynamic reconfiguration techniques available for current families of programmable devices [23], as well as using comparison strategies such as those described later in this work.

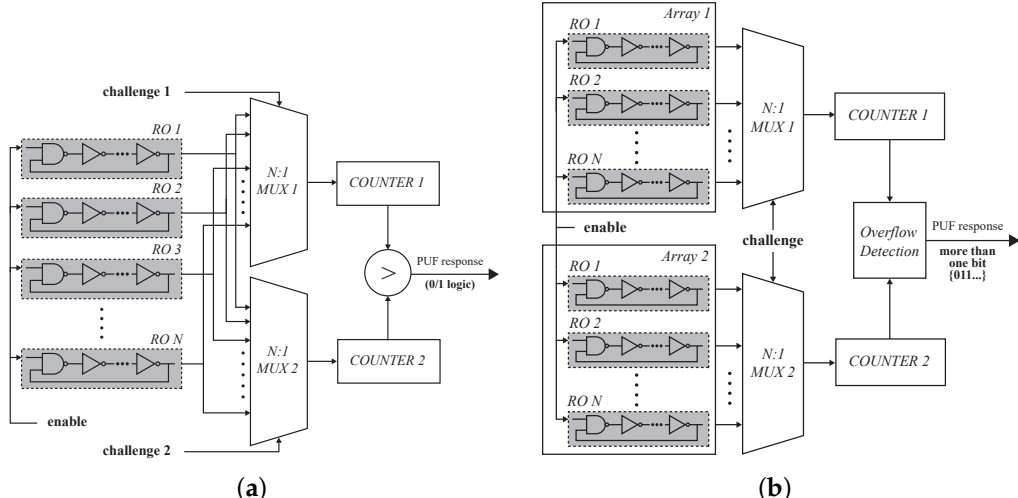

**Figure 1.** Block diagrams of (**a**) conventional RO-PUF in [10], and (**b**) RO-PUF presented in [17].

In addition to the bits-per-area ratio, the main features that define the quality of a PUF are its reliability and uniqueness. Reliability determines the extent to which the PUF response is repeatable throughout the lifetime of the device, while uniqueness establishes its potential to generate an output that is unique and identifies univocally to this device. Both magnitudes can be quantified for a given PUF by evaluating the Hamming distances between the codes resulting from the repeated application of the challenges sequence to the same PUF (intra-Hamming distance, *HDintra*) and to other replicas of it placed in other locations on the same programmable device or in the same location on different programmable devices (inter-Hamming distance, *HDinter*), respectively. The optimal value for the *HDinter* metric is 50%. The desirable value of *HDintra* is 0%, which means that the response that produces a given PUF implementation is always the same.

Unfortunately, this last objective is hardly achieved as a consequence of different sources of noise in the device as well as small changes in the operating conditions (mainly operating temperature and voltage), which cause the PUF response to vary slightly in successive applications of the same sequence of challenges. Under these circumstances, to improve the repeatability of the PUF output for a sequence of challenges so that it can be used in device authentication applications, a helper data algorithm must be applied to achieve the required reliability. HDAs typically consider three stages: bit selection, Error-Correcting Codes (ECCs), and entropy compression [24].

Bit selection is essential to ensure an acceptable starting value in the reliability of a PUF. How this selection is carried out depends on the type of PUF. In PUFs that only use the sign bit of the comparisons, it basically consists of choosing the RO pairs that are involved in each comparison. The approach followed in [10] is to select ROs for each comparison of eight possible candidates, choosing those with the largest frequency differences to increase the robustness of the PUF against environmental variations and noise. In [11], the ROs are placed as close as possible in a 2D matrix, and two adjacent ROs are used in each comparison. A configurable Ring Oscillator PUF that allows choosing the most suitable stages in each RO is described in [12]. Other techniques to improve the reliability of PUFs are based on generating enable signals to activate only the ROs involved in the comparisons [13], choosing the most appropriate challenge–response set [14,15], or using a sensor integrated on-chip to select the pairs of ROs based on their performance in a temperature range [16].

In PUFs whose output incorporates more than one bit from each comparison, the appropriate selection of these bits is essential to maximize the quality of the PUF. The subset of metrics used to accomplish this task includes the average stability (*S*) and probability (*P*) as well as the entropy per bit [17,18]. The ideal value for stability is 1, which means that the bit output is reproducible in all responses (reliable). A value of 0.5 in the average

probability ensures that there is no tendency toward a given logical value (no bias). Finally, to corroborate if the PUF output fulfills the uniqueness requirement, two metrics of entropy are evaluated: the intra-entropy (*Hintra*) to evaluate the uniqueness of the PUF output bits within each PUF implementation, and the inter-entropy (*Hinter*) to evaluate the bit uniqueness for each of the RO pairs in different PUF implementations. A maximum entropy (*Hintra* and *Hinter*) equal to 1 guarantees that there is no correlation between the different output bits at each PUF, and there is no correlation between the same bits among different implementations (unique and unpredictable).

The different bit-selection metrics often do not reach their ideal values. In these cases, the other two stages of an HDA can be included to improve the performance of the PUFs so that they can be used as reliable and robust hardware security elements. ECCs improve the reliability of PUFs by reducing the effect of noisy output [25]. The proposed techniques range from the use of a simple scheme (such as a repetition code) or a combination of ECCs that are efficient in terms of resource consumption to more complex ones, which significantly reduce the size of helper data, based on polar coding [26] or nested polar and Wyner–Ziv coding [27]. Other pre- and post-processing techniques have been described in the literature to improve the quality of identifier and secret key generation schemes, although in most cases, they are difficult to implement on resource-limited embedded systems. On the other hand, compression can be used to increase the entropy in the PUF response in order to decrease correlations or bias that lead to information leakage that can be exploited by an attacker to reduce the search space to obtain the secret. A hash function can be used to improve the entropy of the PUF response and minimize leaks by compressing the original output into a shorter one [28]. It is worth mentioning that the usage of both ECC and entropy compression increases the number of bits required from the PUF output to obtain a key of a given length.

A test structure to analyze different strategies for the design of RO-PUFs on Xilinx programmable devices was recently described in [19] by the authors. In that study, different alternatives were considered for the number of stages of the ROs, the generation of the challenges sequence, the choice of the size of counters, and the selection of the output bits of the PUF. In particular, the two bit selection options described above (sign bit or bits chosen from the slower counter) were compared, observing that each of them is appropriate depending on the relative location of the two compared ROs. In addition, with the idea of optimizing the response time of the PUF, tests were carried out using counter sizes between 14 and 16 bits, showing a similar behavior in terms of the metrics used to evaluate the reliability and uniqueness of the different configurations analyzed. Based on the results of this previous study and incorporating some of the proposals that appear in the literature, the following section describes the structure, building blocks, and functionality of a new RO-PUF implemented as a configurable IP module with the following features:

- Compact: optimizes the use of logic elements available in the Configurable Logic Blocks (CLBs) of Xilinx 7-Series programmable devices.
- Efficient: in terms of cost (bits per number of resources), by simultaneously comparing two pairs of ROs and extracting two bits from each comparison, and regarding operation speed (bits per unit of time), by allowing the effective counter size to be adjusted depending on the target device.
- Functional: incorporates in the design a challenges generation mechanism, a bit selection scheme, and an output memory to store the PUF response.
- Reusable: provided as a highly configurable IP module, with a standard connection interface and drivers that make its use easy from the general-purpose processor of an embedded system.

## 3. RO-PUF IP Module Design and Implementation

The internal structure of the proposed RO-PUF is shown in Figure 2. As with other PUFs based on ring oscillators, its operation essentially consists of comparing the oscillation frequencies of pairs of elements selected among those available in a bank of ROs

(*ro_bnk*). To do this, the output signals of the two ROs being compared incrementally increase the values of two counters, so that when one of them overflows, the counting process is interrupted to identify the faster counter (which determines the sign bit) and acquire the value of the slower counter (from which the rest of the bits for the PUF output corresponding to this comparison will be extracted). The output of the RO-PUF is a bitstream conformed by the concatenation of the bits selected for each of the comparisons after the complete challenges sequence has been applied.

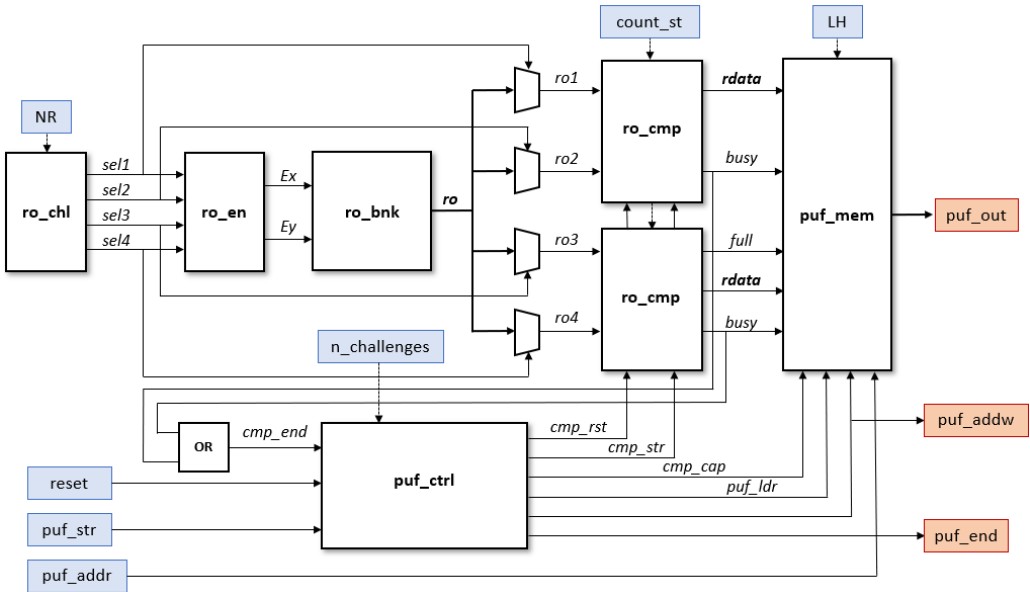

**Figure 2.** Block diagram of the proposed RO-PUF.

However, in our contribution, two simultaneous comparisons will be carried out in parallel, taking advantage of the two different behaviors identified in [19], depending on whether the comparison is made between two ROs implemented in LookUp Tables (LUTs) placed in the same position of different CLBs or between ROs implemented in LUTs placed in different positions within the same or a different CLB, thus doubling the bit generation rate in the PUF response. The selection and enabling signals for the two pairs of ROs to be compared in each comparison cycle are provided, respectively, by the challenges generation (*ro_chl*) and enable (*ro_en*) blocks. On the other hand, the information provided by the two comparison blocks (*ro_cmp*) constitutes the input to the PUF output block (*puf_mem*). This block first chooses the most suitable bits for each of the two mentioned comparisons: in the first case, the sign bit plus a bit from the counter associated to the slower RO that has adequate values of *S*, *P*, *Hintra* and *Hinter*, and in the second, two bits of the counter incremented by the slower RO that meet the same condition. Finally, the PUF output, consisting of a bitstream generated by the concatenation of the bits selected when applying the challenges sequence, is structured in 32-bit registers and stored in the PUF internal memory, which can be read thought the AXI4 interface. The implementation details of each of the building blocks are described in the following subsections.

### 3.1. RO-PUF Building Blocks
#### 3.1.1. RO bank (*ro_bnk*)

The main component of the RO-PUF is a matrix of $Nx$ columns by $Ny$ rows of CLBs, in which each CLB implements four four-stage ROs. As illustrated in the schematic of Figure 3a, three stages of each RO correspond to logic inverters, while the fourth is a NAND gate whose objective is twofold: it closes the feedback loop of the ring oscillator and receives the row and column enable inputs. The Xilinx Series-7 and Zynq-7000 CLBs include eight LUTs, each of which can implement two independent Boolean functions of five inputs

or less [29]. Therefore, by using appropriate placement directives in the VHDL description, it is possible to place the four ROs in the same CLB, taking full advantage of the logical resources of the programmable device.

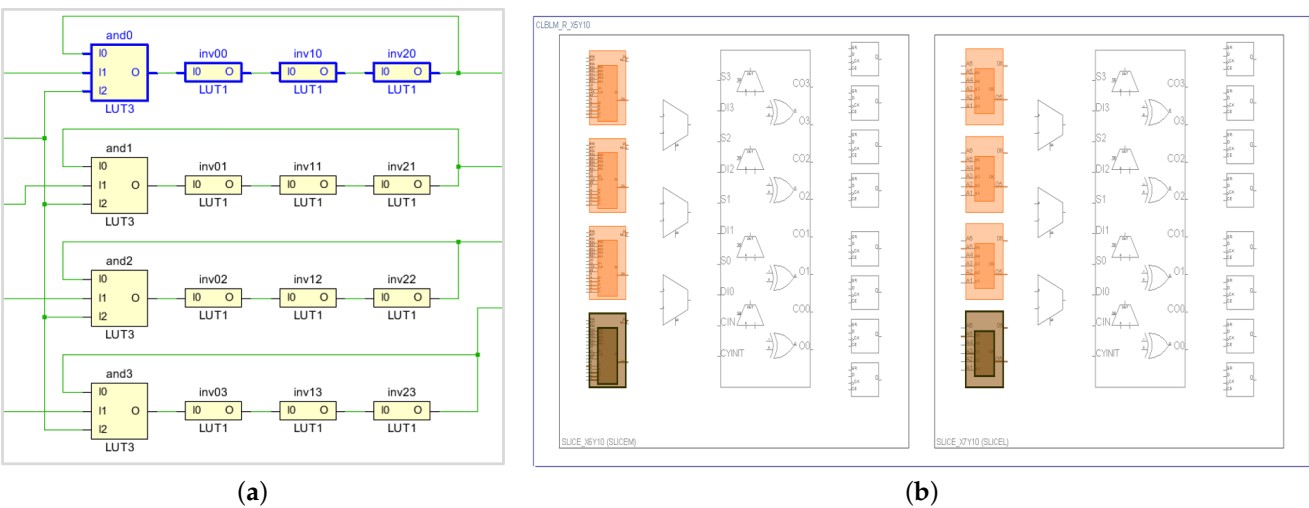

(**a**)                      (**b**)

**Figure 3.** Four four-stage ROs implemented on a CLB: Schematic (**a**) and device (**b**) representation.

Another question arises regarding the location of the ROs within the CLB. It is usually argued that the two ROs to be compared in a PUF must have identical layouts, so that the difference in their oscillation frequencies is only due to variations in the manufacturing process. Information concerning the internal layout of programmable devices is not usually available to designers, but it is reasonable to assume that the same geometric pattern is maintained in CLBs with the same functionality. Considering that the left and right slices of certain CLBs provide different functionalities and, therefore, their layouts must differ, in the proposed design, location constraints are used to force a horizontal layout (shown in Figure 3b) in order to obtain closer oscillation frequencies between the ROs.

### 3.1.2. Challenges Generator (*ro_chl*)

The challenges generator block provides the challenges sequence that determines the two pairs of ROs to be compared in each comparison cycle. Any pair of ROs can be compared, including those located on the same CLB. The block provides four outputs ($sel1, sel2, sel3, sel4$) that are connected to the enable signal generator block and to the control inputs of the multiplexers that select the ROs that will act as clock inputs in the comparison blocks. The $sel1$ signal is generated by a counter, which increments by one ($sel1 = sel1 + 1$) on each comparison cycle. The other selection signals depend on $sel1$ according to Equation (1),

$$sel2 = sel1 + 1 + s\_inc \times 4; \quad sel3 = sel1 + 2; \quad sel4 = sel1 + 6 + s\_inc \times 4 \quad (1)$$

where $s\_inc$ allows us to define the distance, in terms of number of CLBs, between ROs.

The $sel1$ and $sel2$ signals determine the two ROs involved in the first comparison. They select ROs implemented in LUTs located at different positions in the same or contiguous CLBs (if $s\_inc = 0$) or in two different CLBs (for $s\_inc$ in $[1, Nx \times Ny - 1]$). On the other hand, $sel3$ and $sel4$ control the ROs participating in the other simultaneous comparison. The elements selected by these signals correspond to ROs implemented in LUTs located at the same position of two CLBs that are contiguous ($s\_inc = 0$) or separated by a certain distance ($s\_inc \neq 0$).

To provide flexibility to choose different configurations, the proposed RO-PUF includes an online mechanism to select whether the two simultaneous comparisons are made between the closest or farthest ROs of each type within the RO-bank. Depending on the value of the NR (Nearby/Remote) option, in the first case, a null value is set for $s\_inc$,

while in the second case, an internally calculated value is used based on the parameters $Nx$ and $Ny$ that determine the size of the PUF RO-bank.

### 3.1.3. Enable-Signals Generator (*ro_en*)

With the goal of minimizing the activity of the components of the RO-block to reduce energy consumption and avoid mutual influences between them, only the four ROs corresponding to the applied challenge are activated in each comparison cycle. The enable signal generation block (*ro_en*) is responsible for activating row (*Ey*) and column (*Ex*) enable signals, which close the feedback loop of the four ROs indicated by *sel*1–*sel*4. To simplify the implementation of this block, only values of $Nx$ that are powers of two are allowed in the PUF design.

### 3.1.4. Comparison Block (*ro_cmp*)

Two identical comparison blocks (*ro_cmp*) are included in the PUF to perform the two simultaneous comparisons that provide the response corresponding to a challenge. Each of these blocks contains two counters, which use as count signals the output of the two selected ROs, as well as the logic required to stop the operation of the other counter when one of them reaches the maximum value.

The maximum size of the counters is fixed through a parameter established when synthesizing and implementing the design. However, with the idea of minimizing the PUF response time and optimizing it for different target devices, the *count_st* input shown in the block diagram of Figure 2 can be used to define an effective length less than the maximum in each invocation of the PUF.

The comparison cycle starts simultaneously in both blocks, when the *cmp_str* signal is activated by the PUF control block, and it ends when the *busy* outputs of both blocks go down to 0 to indicate that one of the two counters has reached its maximum value. Then, the signals that identify the faster counter in one of the blocks and the output of the slower counter in both blocks are accessible to the input of the last stage in the block diagram of the design.

### 3.1.5. PUF Output Block (*puf_mem*)

The functionality of the PUF output stage (*puf_mem*) is twofold. On the one hand, it selects the bits that will be part of the PUF response for each challenge. On the other hand, as the application of the sequence of challenges progresses, it is in charge of structuring the successive responses in 32-bit registers and storing them in a memory, from which the PUF output will be read once its operation has been completed.

The selected bits depend on the type of ROs being compared as well as on a configuration option (Lower/Higher) defined at run-time by the user. For comparisons between ROs implemented in LUTs located in different positions of the CLB, bits 6 and 7 (for notation purposes, we call bit 0 the sign bit, and the rest of the bits are named in ascending order, bit 1 being the MSB of the counter value, as so-called in other works in the literature [17–19]) (for the Lower option) or bits 7 and 8 (for the Higher option) of the slower counter are chosen to form part of the PUF output. On the other hand, in comparisons between ROs implemented in LUTs located in the same position of different CLBs, their contribution to the output of the PUF will consist of the sign bit in combination with bit 7 (Lower option) or 8 (Higher option) of the slower counter.

The four bits selected in each comparison cycle are sent to a 32-bit shift register, which is in charge of organizing the PUF output bitstream in registers of this size and storing them in consecutive locations in the PUF memory, which are implemented using Block RAM (BRAM) in the programmable device. The PUF output can be accessed from outside the design using the address and data buses associated with this memory.

### 3.1.6. Control Block (*puf_ctrl*)

The control signals necessary to coordinate and sequence the operation of the different blocks are provided by the *puf_ctrl* block. The VHDL description of this block includes two types of components: a Finite State Machine (FSM) to generate the signals controlling the comparison cycles and a series of processes to generate the signals defining the different operation phases and controlling the access to the PUF memory.

The FSM receives two external inputs: *n_challenges*, which defines the number of challenges used in the PUF invocation (i.e., PUF-length/4), and *puf_str*, which sets the start of the PUF operation, as well as the internally generated *cmp_end* signal indicating the completion of the two comparisons. It provides as output the *cmp_rst* and *cmp_start* signals, to initialize and start the comparisons, respectively, and the *cmp_cap* signal, to capture the bits selected in the two simultaneous comparisons.

Figure 4 shows the FSM state diagram. FSM operation starts from an IDLE state in which the three output signals (*cmp_rst*, *cmp_str*, and *cmp_cap*) are deactivated by setting them to 0. When the *puf_str* signal goes high, the FSM goes to the CMP_RESET state, and *cmp_rst* is activated to reset the counters of the two comparison blocks. After one clock cycle, the FSM goes directly to the CMP_DLY state and deactivates *cmp_rst*, and after another clock cycle, it goes to the CMP_START state and activates *cmp_str* to start the operation of both comparison blocks. The FSM waits in the CMP_CYCLE state until both comparisons are complete and the *cmp_end* input is set. When this happen, it goes to the CMP_CAPTURE state and activates *cmp_cap* to capture the four bits that are sent to the shift register to be part of the PUF output. In the next clock cycle, the FSM returns again to the IDLE state, waiting for the start of a new comparison cycle.

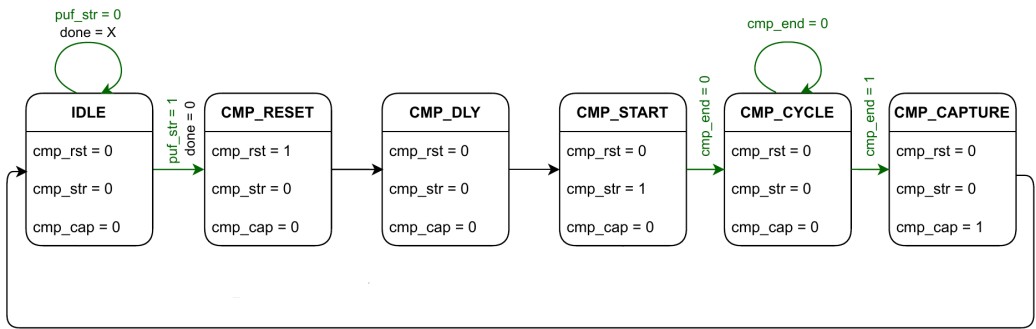

**Figure 4.** State diagram of the FSM included in the control block.

Each time the *cmp_cap* signal goes high, indicating that a comparison cycle has finished, a counter is incremented to record the number of challenges evaluated. When eight challenges have been completed, the *puf_ldr* signal is activated to store the content of the shift register in the PUF memory location indicated by *puf_wa*, and its value is increased by one. Finally, when the number of evaluated challenges is equal to the value defined by the *n_challenges* input, the *done* output signal is activated to indicate that the PUF operation has finished.

### 3.2. IP Encapsulation and Test System Integration

The PUF design has been encapsulated as a configurable IP module with an Advanced Extensible Interface (AXI) bus for interconnection with general-purpose processors. The selected protocol, AXI4-Lite, allows for low resource implementation, which is especially suitable for connecting processors with memory-mapped low- or medium-speed peripherals. The interface uses three channels for write operations (address, data, and response) and two more for read operations (address and data), with 32 or 64 bits for width of data.

The inputs and outputs represented in Figure 2 in blue and red, respectively, are connected to four 32-bit registers following the bit association scheme shown in Figure 5. The input register CONTROL is used to provide the PUF with the number of challenges

(*n_challenges*), the counter-stop mask (*count_st*), and configuration options (*LH* and *NR*), as well as to send the PUF initialization (*reset*) and operation start (*puf_str*) signals. All fields are fixed lengths except for the first one, which depends on the size established when implementing the PUF. PUFADDR is also an input register that is used to access the PUF memory once its operation has finished. The maximum number of bits to represent the read memory addresses (*puf_addr*) is adequately adjusted when the design is synthesized, depending on the length of the PUF response and, consequently, the number of memory cells required to store it. The DATAOUT output register contains three fields. ID is a user-defined identifier, which can be set by the designer for debugging or verification purposes when he/she instances the IP into a higher-level design. On the other hand, *puf_end* is a signal that indicates the PUF has finished its operation, while *puf_addw* contains the address of the last memory position containing the PUF output, so allowing the user to corroborate that it has the expected length. Finally, the PUFOUT register provides in the 32-bit *puf_out* field the content of the PUF memory location addressed by *puf_addr*.

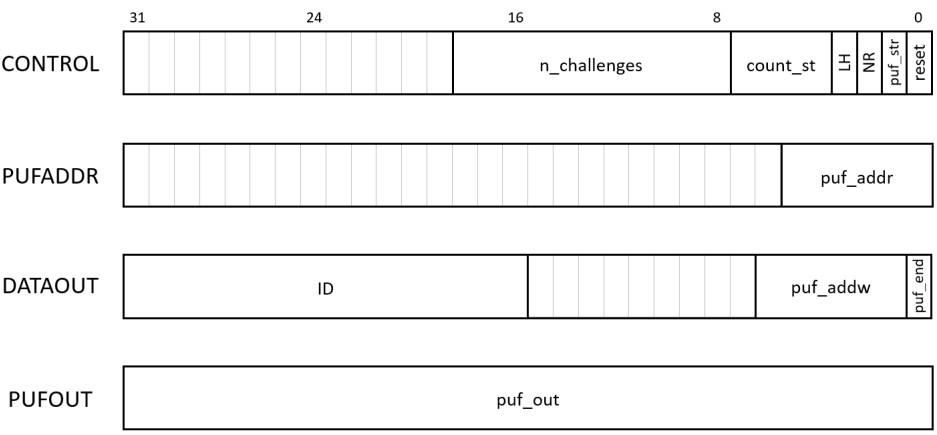

**Figure 5.** Input and output IP module registers.

To optimize its implementation and facilitate its use in different applications, the design of the PUF has been extensively parameterized. Some of these parameters can be defined by designers through a Graphical User Interface (GUI) when using Vivado's IP integrator tool to incorporate the PUF into their design. Specifically, the set of parameters that can be defined through the PUF GUI includes the number of rows ($Ny$) and columns ($Nx$) of adjacent CLBs that make up the RO-bank, its location within the programmable device ($Xo$, $Yo$ coordinates), the maximum number of bits of the counters used to compare the frequencies of the ROs ($Nbc$), and the identifier associated with the PUF ($ID$).

Test System

Programmable SoCs that combine a Processor System (PS) and Programmable Logic (PL) in an integrated circuit have become excellent platforms for the prototyping and implementation of small series of devices for validation and performance analysis of new designs. They put together the flexibility provided by software with the efficiency gained by implementing part of the system on dedicated hardware specially tailored to a given application. Taking advantage of these features, a test system has been implemented in the Xilinx Zynq-7000 SoC device available on the Pynq-Z2 development board to facilitate the validation and characterization of the proposed PUF through a series of routines encoded in C language and executed on one of the ARM cores provided by the device.

The test system instantiates 10 identical RO-PUF IP modules, each with 8 rows and 15 columns of CLBS (containing 480 ROs) and a maximum counter size of 15 bits. The locations of the PUF RO-banks are distributed in the different clock zones present in the device. The remaining components of each PUF are placed in resources belonging to the same clock zone and close to the RO-bank with the help of 'pblock' directives. Figure 6 shows a sym-

bolic representation of the programmable device, in which the distribution of the different PUFs can be observed. Orange cells correspond to the RO-banks whose positions were established when the PUFs were instantiated. The purple boxes mark the zones defined by the 'pblock' directives to locate the other components of each PUF. Finally, the cells in green show the device resources that are fully or partially used.

Each of the PUFs occupies 1862 LUTs (3.50% of the resources in the device) where 960 LUTs (1.80% ) are used by the matrix of ROs. It also consumes 365 (0.34%) Slice Registers, 256 (0.96%) F7 Muxes, 119 (0.89%) F8 Muxes, and 0.5 (0.36%) BRAMS. The amount of resources consumed by the complete test system, including those implementing the AXI4 infrastructure required to connect the PUFs and the PS, are 19289 LUTs, 4270 Slice Registers, 2560 F7 Muxes, 1190 F8 Muxes, and 5 BRAMS.

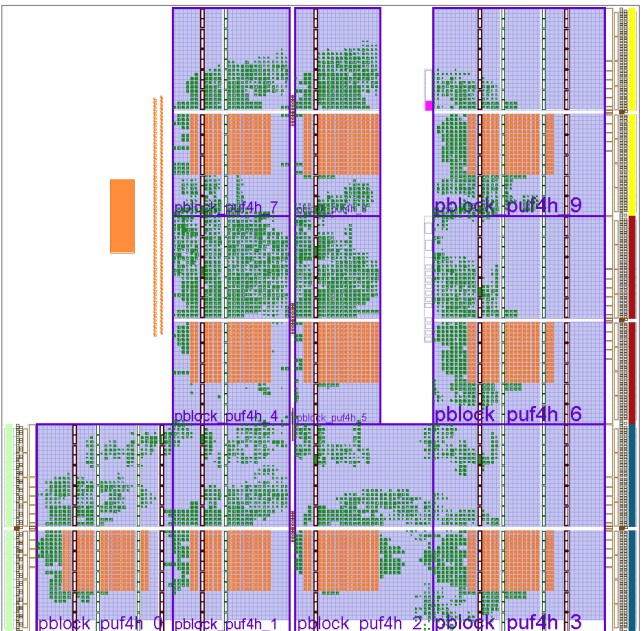

**Figure 6.** Device view of the test system implementation.

## 4. RO-PUF Characterization

The objective of the RO-PUF characterization task is twofold: on the one hand, verify that the bits selected in each comparison according to the possible options present adequate values of stability, probability, and entropy; on the other hand, obtain a series of metrics that allow evaluating how the setting of the different configuration options affects the PUF reliability and uniqueness.

To meet this dual objective, an extensive battery of tests has been developed taking advantage of the Python Productivity for Zynq (PYNQ) environment available for Pynq-Z2 boards [30]. It provides a Python framework on an embedded Linux operating system, which simplifies the interaction between the hardware and software components of an embedded system. For efficiency reasons, in this work, we use the C-API available in [31], which provides the same functionality through a set of C routines that can be compiled to generate executable code. This API includes functions to handle the hardware elements integrated into PYNQ as a hardware equivalency to software libraries.

A series of specific test functions have been coded in C in order to repeatedly invoke the different PUFs instantiated in the test system and capture the corresponding output data. When these tests are launched, the number of challenges, the number of tests, i.e., the number of PUF calls, and the debug level can be configured by the user. Different strategies can also be applied by combining configuration options for the selection of lower or higher bits, nearby or remote ROs, and the effective size of counters. Once the tests are

run through command-line or shell scripts, the output data are captured and stored in files for posterior processing.

To carry out the study, 10 development boards were used, each one implementing the test system with 10 PUFs, which means a total of 100 different RO-PUFs. In all cases, the four configuration options that arise when considering the relative position of the ROs involved and the bits selected in each comparison were analyzed. The number of calls to each PUF and the effective length of the counters varied depending on the specific objective of the test performed. Once the data are captured, a set of MATLAB scripts and functions are used to calculate the different metrics that allow evaluating the quality of the proposed PUF.

### 4.1. Bit-Selection Analysis

Figure 7 shows the average stability ($S$), probability ($P$), and entropy (*Hintra* and *Hinter*) per bit, which are calculated when a complete sequence of 480 challenges is applied 1000 consecutive times to each of the 100 PUF, and the obtained responses, each of them composed of a stream of 1920 bits, are captured and processed. The data in each bar graph are grouped according to the four alternatives that arise when considering the possible combinations of the LH and NR configuration options. In all cases, label 1 corresponds to the sign bit of the second comparison, while the bits represented by the other three labels depend on the specific configuration: label 2 is bit 6 (L) or 7 (H) of comparisons between ROs implemented in LUTs that occupy the same positions in different CLBs; labels 3 and 4 correspond to bits 6 and 7 (L) or 7 and 8 (H) when comparing ROs implemented in LUTs located at different positions.

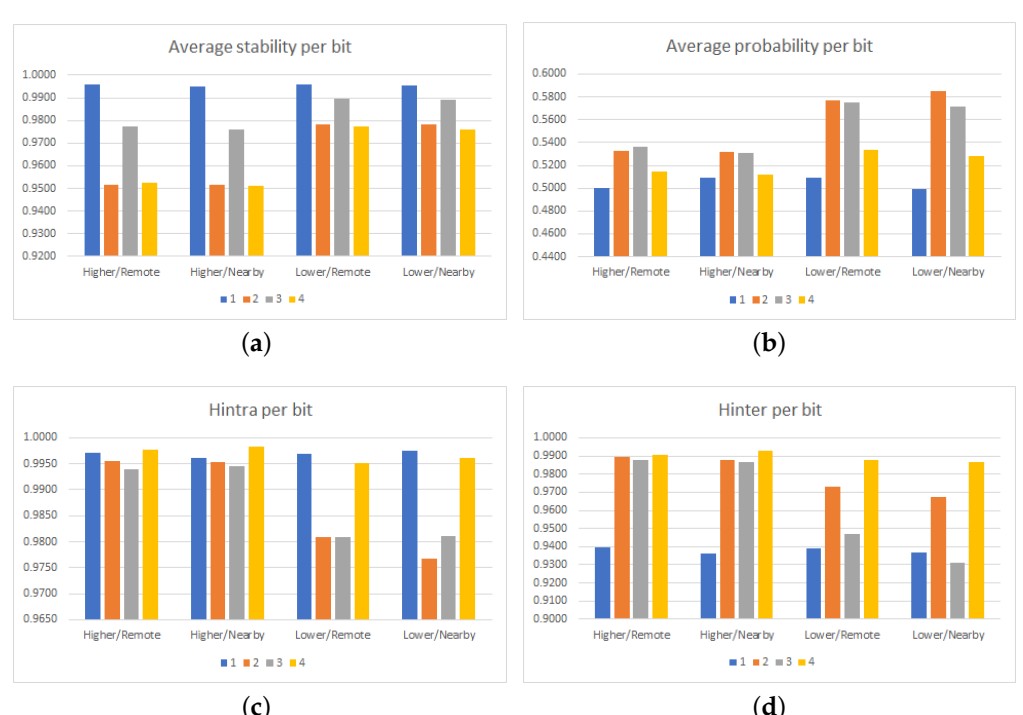

**Figure 7.** Average stability (**a**), probability (**b**), and entropy per bit (**c**, **d**).

As can be seen in the graphs, the configurations that use lower bits of the counters (L) present greater stability and probability, although their entropy values are lower than those of the configurations that use higher bits (H), which was predictable. As for the relative position of the compared ROs, no significant differences in stability are observed for configurations comparing nearby (N) or remote (R) ROs, although it does seem to affect the other three metrics in some way but without showing a clear trend for any of the bits considered.

Once we verified that the bits selected in the different configurations meet the conditions required to form part of the PUF output, the following sections show the results of the tests carried out to determine their performance in terms of reliability and uniqueness.

### 4.2. PUF Performance Evaluation

As mentioned in the Introduction to this work, *HDintra* and *HDinter* are the metrics commonly used to assess the reliability and uniqueness of a PUF. Therefore, the results obtained when the RO-PUF behavior is evaluated considering different operating conditions are summarized below. Several analyses have been carried out in order to determine: (a) the influence of selecting Lower or Higher bits and Nearby or Remote ROs, to check which strategy provides better performance; (b) the impact of the effective size of the counter to minimize the PUF response time; and (c) the incidence of the number of calls to the PUF to verify the generality of the results obtained.

#### 4.2.1. Selection Strategies

This study was carried out on the same number of PUFs used to perform the bit selection analysis. Data obtained in 1000 successive calls to a total of 100 PUFs, with 480 ROs each and an effective counter size of 14 bits, were processed with the help of a series of MATLAB functions and scripts coded for this specific purpose. The mean *HDinter* values, as well as the mean, minimum, and maximum *HDintra* values, for each PUF of the test system, corresponding to the four possible configurations or comparison strategies, are shown in Figure 8.

| Boards = 10 | | | | | | | | |
|---|---|---|---|---|---|---|---|---|
| **00 - Higher bits/Remote ROs** | | | | | **01 - Higher bits/Nearby ROs** | | | |
| **PUF** | **HDInter** | **HDIntra** | **HDIntra_min** | **HDIntra_max** | **PUF** | **HDInter** | **HDIntra** | **HDIntra_min** | **HDIntra_max** |
| 001 | 43.05 | 3.22 | 2.68 | 4.12 | 001 | 43.53 | 3.11 | 2.52 | 3.76 |
| 002 | 43.42 | 3.08 | 2.59 | 3.75 | 002 | 42.95 | 3.28 | 2.82 | 3.78 |
| 101 | 41.89 | 3.08 | 2.64 | 3.60 | 101 | 42.65 | 3.17 | 2.49 | 3.85 |
| 102 | 41.69 | 3.02 | 2.50 | 3.59 | 102 | 42.39 | 2.98 | 2.59 | 3.71 |
| 011 | 43.08 | 3.18 | 2.63 | 3.90 | 011 | 43.25 | 3.20 | 2.64 | 3.82 |
| 111 | 40.65 | 2.95 | 2.42 | 3.33 | 111 | 40.38 | 3.13 | 2.63 | 4.07 |
| 112 | 41.51 | 3.02 | 2.43 | 3.55 | 112 | 40.82 | 3.13 | 2.89 | 3.55 |
| 021 | 43.25 | 3.13 | 2.70 | 3.88 | 021 | 43.90 | 3.16 | 2.58 | 3.65 |
| 121 | 41.51 | 3.04 | 2.47 | 4.33 | 121 | 40.89 | 3.21 | 2.25 | 5.22 |
| 122 | 43.73 | 2.90 | 2.32 | 3.62 | 122 | 43.52 | 3.21 | 2.63 | 3.90 |
| **All** | **48.94** | **3.06** | **2.32** | **4.33** | **All** | **48.89** | **3.16** | **2.25** | **5.22** |
| **10 - Lower bits/Remote ROs** | | | | | **11 - Lower bits/Nearby ROs** | | | |
| **PUF** | **HDInter** | **HDIntra** | **HDIntra_min** | **HDIntra_max** | **PUF** | **HDInter** | **HDIntra** | **HDIntra_min** | **HDIntra_max** |
| 001 | 40.34 | 1.43 | 0.94 | 1.95 | 001 | 40.27 | 1.51 | 1.40 | 1.73 |
| 002 | 40.80 | 1.46 | 1.28 | 1.59 | 002 | 39.51 | 1.56 | 1.27 | 1.84 |
| 101 | 39.92 | 1.50 | 1.14 | 2.07 | 101 | 39.89 | 1.53 | 1.16 | 1.89 |
| 102 | 39.48 | 1.48 | 1.14 | 2.00 | 102 | 39.02 | 1.42 | 1.19 | 1.64 |
| 011 | 40.43 | 1.52 | 1.29 | 2.01 | 011 | 39.90 | 1.51 | 1.15 | 2.15 |
| 111 | 38.79 | 1.44 | 0.90 | 1.85 | 111 | 38.41 | 1.52 | 1.20 | 1.80 |
| 112 | 39.19 | 1.43 | 1.19 | 1.72 | 112 | 38.05 | 1.50 | 1.18 | 1.77 |
| 021 | 40.58 | 1.48 | 1.20 | 1.80 | 021 | 40.36 | 1.58 | 1.28 | 1.90 |
| 121 | 39.48 | 1.54 | 1.21 | 2.32 | 121 | 38.92 | 1.64 | 1.16 | 2.75 |
| 122 | 41.36 | 1.39 | 0.92 | 1.84 | 122 | 40.30 | 1.56 | 1.01 | 2.01 |
| **All** | **47.93** | **1.47** | **0.90** | **2.32** | **All** | **47.52** | **1.53** | **1.01** | **2.75** |

**Figure 8.** *HDinter* and *HDintra* values for different selection strategies of bits and ROs.

The first 10 rows of each table show the values of *HDinter* and *HDintra* associated to each of the PUFs instantiated in the test system. The value of *HDinter* corresponds, in this case, to the average Hamming distance between the responses of a given PUF and those of the PUFs implemented in the same position in the other nine development boards. The mean, minimum, and maximum values of *HDintra* are calculated as the average, minimum, and maximum, respectively, of the Hamming distances between the successive

responses of the same PUF. The last row in each table collects global values when considering all PUFs. *HDinter* is now calculated as the average of the Hamming distances between the responses of one PUF and those of the other 99 PUFs. *HDintra*, *HDintra_min*, and *HDintra_max* correspond to the mean, minimum, and maximum values of the top 10 rows.

As can be seen from the global values highlighted in Figure 8 (and summarized in Table 1 for the sake of a better comparison), *HDinter* values range from 47.52 (Lower bits/Nearby ROs) to 48.94 (Higher bits/Remote ROs), while the *HDintra* mean ranges from 1.47 (Lower bits/Remote ROs) to 3.16 (Higher bits/Nearby ROs). Considering each of the options separately, the Lower option provides better performance in terms of reliability (lower *HDintra*); however, the *HDinter* is less than 48%. The Higher option allows increasing *HDinter* by more than one point, but at the cost of doubling the *HDintra* value. Different behavior is obtained when selecting the test strategies between Nearby or Remote ROs. In this second case, the mean values of the reliability and robustness indicators of the PUFs are slightly better when Remote ROs are compared.

**Table 1.** *HDinter* and *HDintra* for different strategies.

| Strategy | *HDinter* | *HDintra* | *HDintra_min* | *HDintra_max* |
|---|---|---|---|---|
| Higher/Remote | 48.94 | 3.06 | 2.32 | 4.33 |
| Higher/Nearby | 48.89 | 3.16 | 2.25 | 5.22 |
| Lower/Remote | 47.93 | 1.47 | 0.90 | 2.32 |
| Lower/Nearby | 47.52 | 1.53 | 1.01 | 2.75 |

These data are consistent with the stability and entropy values of the PUF bits obtained in Section 4.1. A lower *HDintra* value implies a lower Bit Error Rate (BER) and therefore better reliability. On the other hand, a value of *HDinter* closer to 50% improves the uniqueness and resistance of the PUF to possible attacks. In this way, by setting the different configuration parameters of the proposed RO-PUF, the selection strategy of bits and/or ROs can be chosen at run-time to establish a reliability/robustness trade-off suitable for a particular application context.

### 4.2.2. Effective Counter Size

In order to analyze the timing response of the PUF, a specific study was carried out to evaluate the impact of the effective size of the counters used when comparing RO pairs. In this case, the metrics were calculated for 30 PUFs distributed on three different boards, using the responses obtained by calling each PUF 1000 times with sequences of 480 challenges. Table 2 shows the *HDinter* and *HDintra* values for PUFs with an effective counter size of 13 and 14 bits, using the Higher option in the upper two rows and Lower option in the lower two rows to compare equivalent bits.

**Table 2.** *HDinter* and *HDintra* for different effective counter size.

| Size | Strategy | *HDinter* | *HDintra* | *HDintra_min* | *HDintra_max* |
|---|---|---|---|---|---|
| 13 | Higher/Remote | 48.14 | 1.46 | 1.03 | 2.33 |
| 13 | Higher/Nearby | 47.91 | 1.52 | 1.15 | 2.74 |
| 14 | Higher/Remote | 48.02 | 1.41 | 0.90 | 2.32 |
| 14 | Higher/Nearby | 47.67 | 1.53 | 1.01 | 2.75 |

The results of this test allow us to verify, as expected, that the behavior of the PUF when using the Higher bits with an effective counter size of 13 bits is similar to that obtained with the Lower bits and an effective counter size of 14 bits, with the clear advantage that in the first case, the response time of the PUF is reduced by half.

The time invested by the PUF to provide the response depends on the characteristics of the programmable device in which it is implemented (family, part, and speed grade), which determine the average oscillation frequency of the ROs as well as the parameters

used when implementing the IP module ($Nx$, $Ny$, $Nbc$), which determine the number of comparisons and the number of clock cycles per comparison. For illustrative purposes, Figure 9 shows the variation in the response times of one of the PUFs included in the test system implemented on the Pynq-Z2 boards when the effective size of the counters varies between 12 and 16 bits. The typical oscillation frequency of the ROs on this development board was about 315 MHz, so the 480-RO PUF spends almost 30 ms to provide the response when using 14-bit counters, but it uses only a little more than half of this time when the effective size of the counters is reduced to 13 bits.

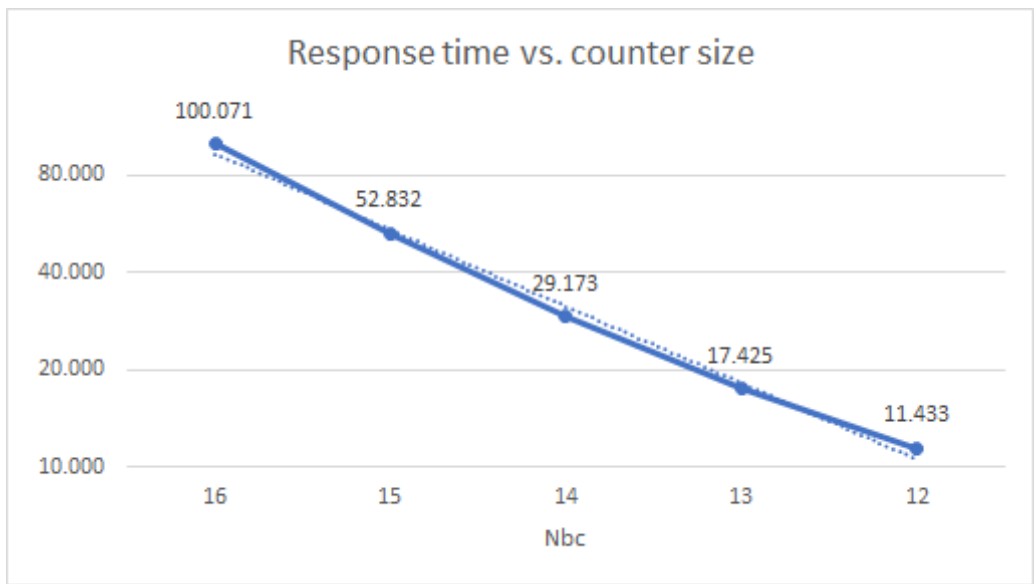

**Figure 9.** Response time (in ms) versus effective counter size (in bits) for one of the 480-RO PUFs implemented in the test system.

### 4.2.3. Number of PUF Calls

To complete the characterization of the proposed PUF, a final test was performed with the idea of verifying the long-term generalization of the results obtained. For this, three successive series of 3000 calls each were made, and data corresponding to 3000, 6000 and 9000 calls were processed to determine the influence of the number of calls on the PUF metrics. To carry out this study, the responses of nine different PUFs distributed in three development boards were considered. Again, PUF responses correspond to the application of a sequence of 480 challenges and provide 1920 bits.

Bit selection metrics (such as those presented in the analysis described in Section 4.1) are shown in Figure 10 for the case in which the PUFs are configured to use the Lower/Nearby options to select bits and ROs, respectively. The results of the other three selection strategies show the same trend, so they have not been included.

As can be seen in the bar charts in the figure, only the average stability per bit shows a very slight decrease (less than a thousandth) when the number of calls to the PUFs doubles and triples, while the variations are practically negligible for the other three metrics.

These data are also consistent with the global values of *HDinter* and *HDintra* shown in Table 3, where it can be seen that the *HDinter* values are similar for all cases and that the average and minimum values of *HDintra* increase a little bit with the number of responses considered.

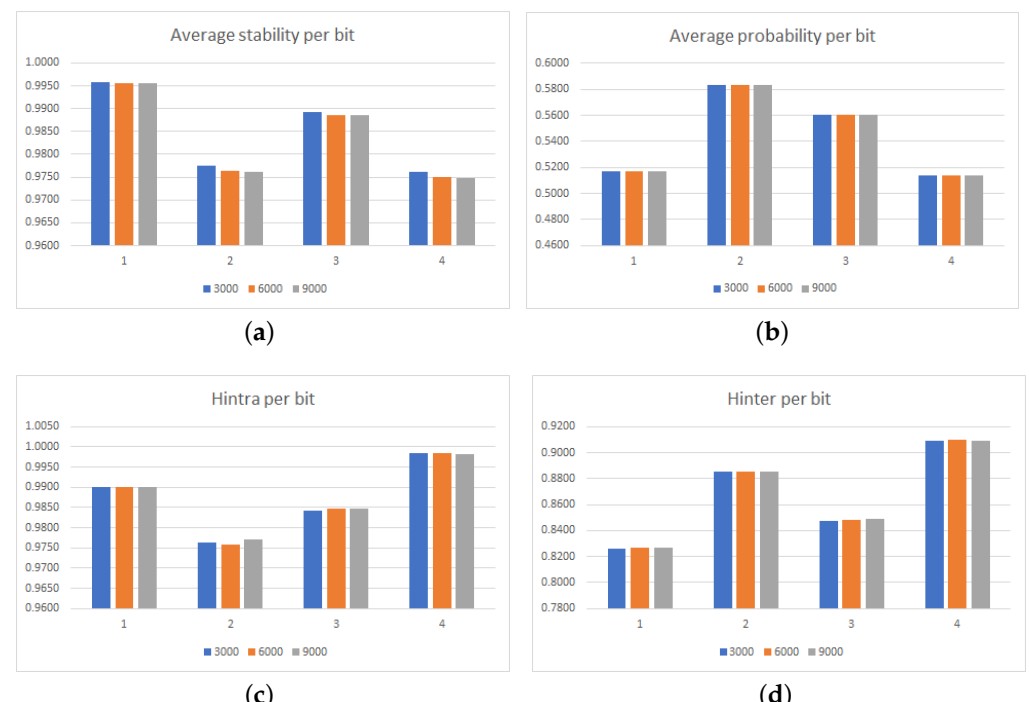

**Figure 10.** Average stability (**a**), probability (**b**), and entropy per bit (**c**,**d**) versus number of calls for the Lower/Nearby selection strategy.

**Table 3.** *HDinter* and *HDintra* versus number of responses for the Lower/Nearby selection strategy.

| N | HDinter | HDintra | HDintra_min | HDintra_max |
|---|---------|---------|-------------|-------------|
| 3000 | 47.03 | 1.54 | 1.09 | 2.12 |
| 6000 | 47.05 | 1.61 | 1.13 | 2.15 |
| 9000 | 47.06 | 1.63 | 1.17 | 2.07 |

## 5. Generation and Recovery of Secret Keys Based on RO-PUFs

To illustrate one of the main applications of PUFs, the use of the proposed RO-PUF to generate and retrieve secret keys is discussed in this section. In this example, a simple ECC consisting of a repetition code scheme is used to deal with the variability in successive PUF responses that has become apparent when evaluating *HDintra* in the previous section.

The scheme to obfuscate and retrieve a secret key using the RO-PUF response and an ECC for a given repetition code, $r$, is shown in Figure 11. In the enrollment (or obfuscation) phase, the secret is extended by replicating $r$ times each bit of the key. Later, the extended key is XOR-ed with the RO-PUF response (which should have the same length of the extended secret, that is, $n \times r$ in which $n$ is the length of the secret key). As a result of the XOR operation between the extended secret key and the RO-PUF response, helper data are obtained. Helper data are non-sensitive data that can be public and stored in any place of the system without being ciphered. In the recovery phase, the secret key is recovered using the helper data generated in the enrollment phase and a new PUF response, which can differ slightly from the one used in the previous phase. The helper data and the new PUF response are XOR-ed to form a new extended secret key, from which the secret key is recovered using an ECC with the same repetition code used in the enrollment. If the PUF response is reliable and robust, only the PUF instance that obfuscated the secret key is the one that can retrieve it, even whether a counterfeit RO-PUF instance has access to the helper data.

To demonstrate the capacity of the proposed RO-PUF to be used as a basic element of an HDA [32], a study was carried out with MATLAB using data obtained from 90 PUFs configured to use the Higher/Nearby selection strategy.

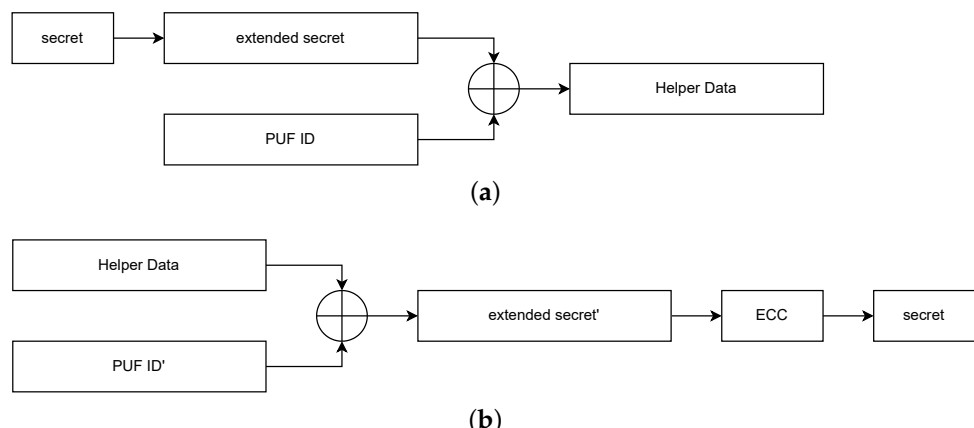

**(a)**

**(b)**

**Figure 11.** HDA scheme to obfuscate (**a**) and retrieve (**b**) a secret key using the RO-PUF.

For each RO-PUF instance, a key was first obfuscated, and then, recovery was attempted using the PUF responses from the same RO-PUF and using the responses from the other 89 RO-PUFs. It was expected that for a given repetition value, only the RO-PUF instance that obfuscated the key would be able to recover it with a desirable False Negative Rate (FNR) of 0, and the rest of the instances would never be able to recover it, which means a False Positive Rate (FPR) of 0. In agreement with the expected results, the calculated FNR for 10 different PUF instances on different boards, after retrieving 1000 times a 128-bit secret key using the 1920 response bits of a PUF, was equal to 0 for an ECC repetition code with $r = 15$, which implies that the secret key could always be retrieved. Likewise, for the same repetition value, the FPR was also always 0, so the secret key could never be retrieved for a PUF instance other than the one used to obfuscate it.

This analysis was also performed by varying the operation conditions of the devices, for which different data sets were collected at different temperatures (0, 14, 28, 42, 56, 70 °C). For each of the data sets collected at a fixed temperature, a key is obfuscated and recovered using the PUF responses of the same RO-PUF and using the PUF responses of the other 89 RO-PUFs for different values of $r$. The experiment was performed by establishing the temperature values over the Pynq boards with a Thermonics temperature source [33] in the lab. Figure 12 shows the minimum value of $r$, where the FNR and FPR are zeros for each of the data sets. It is concluded that it is necessary to use an $r = 19$ for success in all the experiments; that is, the key is always recovered for each of the RO-PUF devices at each of the temperatures considered and without any counterfeit device recovering it.

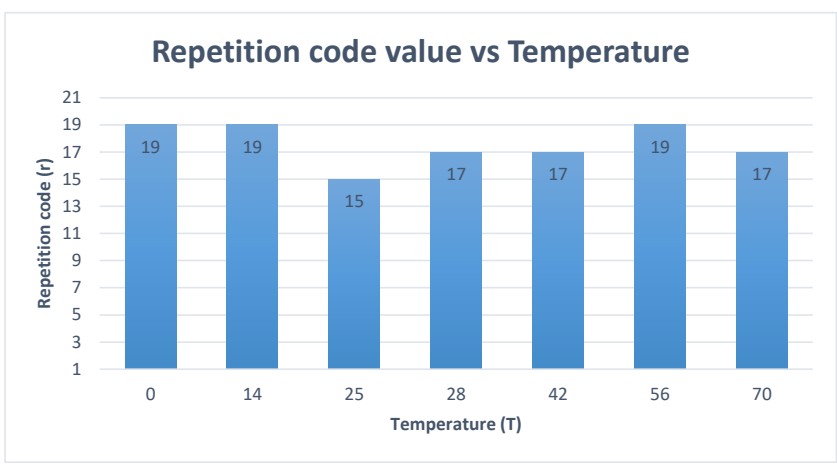

**Figure 12.** Repetition Code Value vs. Temperature (in °C).

## 6. Conclusions

This work addresses the design of a new RO-PUF that is efficient in terms of area and speed of operation as well as its use to generate identifiers and keys for the microelectronic devices in which the PUF is attached/embedded to. The system takes advantage of the resources available in the CLBs of Xilinx 7-Series programmable devices to place four four-stage ROs within one CLB, providing a very compact solution in which the ROs are located in an array of $Nx \times Ny$ CLBs. It also includes a challenges generation mechanism that allows two comparisons to be made in parallel. In both cases, the oscillation frequencies of two Ring Oscillators are compared. One comparison is made between ROs situated at different positions of the CLBs, and the other is made between ROs placed at the same positions of different CLBs. In the first case, two bits of the slower counter are selected to form part of the PUF output, while in the latter, a single bit of the slower counter and the sign bit that indicates which of the two ROs oscillates at a higher frequency are selected. The output of the RO-PUF, formed by the concatenation of the four bits obtained for each challenge, is stored in an internal memory as a sequence of 32-bit registers. This strategy ensures non-correlation between data while allowing the number of output bits obtained in [17,18] to be doubled without significantly increasing either the resources used in the FPGA or the time spent obtaining the PUF response.

The RO-PUF is provided as an IP module, in which an AXI4-Lite interface has been incorporated so that it can be easily integrated into an embedded system. The dimensions of the CLB array, its location within the FPGA fabric, and the maximum size of the counters can be configured before it is implemented. In addition, the PUF functionality can be configured by using the I/O registers mapped into the memory space of the embedded processor. By means of this mechanism, it is possible to determine when invoking the PUF the length of the sequence of challenges and the effective size of the counters as well as to define the strategy to select the position (Nearby/Remote) of the ROs being compared and the location (Lower/Higher) of the bits contributing to the PUF output.

An extensive set of tests has been performed to verify design decisions and characterize the quality of the PUF in terms of the reliability and uniqueness of the outputs it provides. The results of this study show that the different configuration options allow for establishing different reliability/robustness trade-offs as well as optimizing the response time of the PUF depending on the target device in which it is implemented. In addition, using an ECC with a repetition code equal to 15, the feasibility of the PUF as a basic element of an HDA used to generate and recover 128-bit keys with null values of FPR and FNR has been verified.

Finally, although a preliminary characterization of the PUF against changes in temperature has been carried out, a more complete characterization against variations in other operating conditions, such as supply voltage and aging, together with the incorporation of selection techniques that allow increasing the reliability of the PUF, will be some of the tasks that we plan to address as a continuation of this work.

**Author Contributions:** All authors have actively participated in the planning and development of this work. They also collaborated in extensive tests for the characterization of PUFs and in the writing of the paper. M.C.M.-R. programmed the scripts to automate the processing data from PUF responses, L.F.R.-M. performed the verification of both the IP module design and the test system implementation. E.C.-R. prepared the visual support of the published work. S.S.-S. provided the design methodology for the RO-PUF and developed the test battery for its characterization. P.B. coordinated the funding acquisition to support the activities leading to this publication. All authors have read and agreed to the published version of the manuscript.

**Funding:** This research was supported in part by the SPIRS Project with Grant Agreement No. 952622 under the EU H2020 research and innovation programme and the ARES Project PID2020-116664RB-100 funded by MCIN/AEI/10.13039/501100011033 and the EU NextGenerationEU/PRTR. M.C.M.R. holds a Postdoc fellowship from the Andalusia Government with support from PO FSE of EU. E.C.R. is supported by the FPU20/03008 predoc grant from the Spanish government.

**Institutional Review Board Statement:** Not applicable.

**Informed Consent Statement:** Not applicable.

**Data Availability Statement:** Not applicable.

**Conflicts of Interest:** The authors declare no conflict of interest.

## Abbreviations

The following abbreviations are used in this manuscript:

| | |
|---|---|
| ASIC | Application-Specific Integrated Circuit |
| AXI | Advanced Extensible Interface |
| BER | Bit Error Rate |
| BRAM | Block Random-Access Memory |
| CLBs | Configurable Logic Block |
| DRAM | Dynamic Random-Access Memory |
| ECC | Error-Correcting Code |
| FNR | False Negative Rate |
| FPGA | Field-Programmable Gate Array |
| FPR | False Positive Rate |
| FSM | Finite State Machine |
| GUI | Graphical User Interface |
| HDA | Helper Data Algorithm |
| IoT | Internet of Things |
| IP | Intellectual Property |
| LUT | LookUp Table |
| PL | Programmable Logic |
| PS | Processor System |
| PUF | Physical Unclonable Function |
| PYNQ | Python Productivity for Zynq |
| RO | Ring Oscillator |
| SoC | System on Chip |
| SRAM | Static Random-Access Memory |

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
