# Peer review of "Efficient RO-PUF for Generation of Identifiers and Keys in Resource-Constrained Embedded Systems"

_cryptography, doi:10.3390/cryptography6040051_

Round 1

Reviewer 1 Report

This paper presents a Ring Oscillator Physically Unclonable Function for FPGAs solution based on a large state of the art. The paper does not claim much novelty in itself but has the merit to describe a very practical solution and presenting a survey of alternatives. Effort is given to validate this solution. 

 General remarks:

- I missed discussion on the bitrate of your solution, or at least how fast one can retrieve a key in the HDA section

- I have some concerns about the stability of of RO-PUF in FPGAs against voltage change, aging, and bitstream versions (routing is not fixed and may change RO frequency). This concern is not unique to your contribution, but I would have expected you to validate it better since the strong point of your paper is to make it into a practical solution. In particular, it seems possible to alter the internal voltage of the PYNQ-Z2 (vcc1v0) without too much effort. Aging would be more challenging. For vivado, fixing the routes might help a bit.

Other remarks:

page 2:

l.54: I have some doubt that implementing a PUF on an FPGA would be competitive with existing microcontrollers in terms of cost and energy efficiency.

page 6: 

- Block ro_sel is missing from figure 2 (or you need to update your text)

- l.259: ref needs to be fixed

- l.263: This is inaccurate. In 7 devices, each LUT can implement two function of 5 inputs or less

page 8:

-l. 301: competitions -> comparisons

-ro_cmp: The counters are fed by two different clocks, and then I assume the logic that treats the result uses yet another clock. What precautions do you have in place to avoid metastability and hazards ?

-l.323-331: The notation of bit 1 as MSB and 8 as LSB is counter-intuitive to me, and explanation of this notation could occur earlier. Also, do you mean that bits >8 are never used ?

page 10:

- Having an address register for access PUF output seems suboptimal to me. You could just allocate a larger address segment for the IP and use some of the address LSBs...

page 11:

-l.424: YOu may want to use BRAM for consistency 

page 13:

- I do not understand the interest of showing both figure 8 and Table 1 (later being a subset of the first). Personally, I would rather drop figure 8, or at least highlight the numbers that are particularly interesting.

page 16-17:

- You need to add temperature unit (I suppose celsius?)

page 17:

- Somehow, the conclusion is harder to read than the rest of the text

- l602: "In one of them, information is extracted from the sign bit, and in the other from the module of the difference in oscillation frequencies between two ROs, which ensures the non-correlation between the data" -> This is hard to read.
